

# Sequence variability of *Chrysanthemum stunt viroid* in different chrysanthemum cultivars

Hoseong Choi[1,*], Yeonhwa Jo[1,*], Ju-Yeon Yoon[2], Seung-Kook Choi[2] and Won Kyong Cho[1]

[1] Department of Agricultural Biotechnology, College of Agriculture and Life Sciences, Seoul National University, Seoul, Republic of Korea
[2] Virology Unit, Department of Horticultural Environment, National Institute of Horticultural and Herbal Science, RDA, Wan-Ju, Republic of Korea
* These authors contributed equally to this work.

## ABSTRACT

Viroids are the smallest infectious agents, and their genomes consist of a short single strand of RNA that does not encode any protein. *Chrysanthemum stunt viroid* (CSVd), a member of the family *Pospiviroidae*, causes chrysanthemum stunt disease. Here, we report the genomic variations of CSVd to understand the sequence variability of CSVd in different chrysanthemum cultivars. We randomly sampled 36 different chrysanthemum cultivars and examined the infection of CSVd in each cultivar by reverse transcription polymerase chain reaction (RT-PCR). Eleven cultivars were infected by CSVd. Cloning followed by Sanger sequencing successfully identified a total of 271 CSVd genomes derived from 12 plants from 11 cultivars. They were further classified into 105 CSVd variants. Each single chrysanthemum plant had a different set of CSVd variants. Moreover, different single plants from the same cultivar had different sets of CSVd variants but identical consensus genome sequences. A phylogenetic tree using 12 consensus genome sequences revealed three groups of CSVd genomes, while six different groups were defined by the phylogenetic analysis using 105 variants. Based on the consensus CSVd genome, by combining all variant sequences, we identified 99 single-nucleotide variations (SNVs) as well as three nucleotide positions showing high mutation rates. Although 99 SNVs were identified, most CSVd genomes in this study were derived from variant 1, which is identical to known CSVd SK1 showing pathogenicity.

Corresponding author
Won Kyong Cho,
wonkyong@gmail.com

## INTRODUCTION

Viroids are the smallest pathogens that infect plant species (*Diener, 1974*), and the genomes of viroids are composed of a circular single strand of RNA that does not encode any protein (*Tabler & Tsagris, 2004*). Viroid RNAs range in size from 246 to 401 nucleotides (nt) (*Ding, 2010*). So far, more than 30 viroid species have been identified in a wide range of plants, and they have been further divided into two families: *Pospiviroidae* and *Avsunviroidae* (*Di Serio et al., 2014*). Five genera, *Apscaviroid*, *Cocadviroid*, *Coleviroid*, *Hostuviroid*, and *Pospiviroid*, are members of the family *Pospiviroidae*, while three genera, *Avsunviroid*,

*Elaviroid*, and *Pelamoviroid*, are members of the family *Avsunviroidae*. The viroids in the family *Pospiviroidae* with a rod-like conformation replicate in the nucleus, whereas the viroids in the family *Avsunviroidae* displaying highly branched structures with self-cleaving ribozymes replicate in the chloroplast (*Ding, 2009*).

The chrysanthemum (*Dendranthema X grandiflorum*) is famous for its flowers, and numerous cultivars have been developed and cultivated worldwide. Due to the clonal propagation of chrysanthemum cultivars, such as through cutting, the rates of viruses and viroids in cultivated chrysanthemums are very high, resulting in a severe reduction of the quality and quantity in chrysanthemum flower production (*Chung et al., 2005*). So far, nine viruses and two viroids that infect chrysanthemum species have been identified. Interestingly, the chrysanthemum is susceptible to two different viroids: *Chrysanthemum stunt viroid* (CSVd) and *Chrysanthemum chlorotic mottle viroid* (CChMVd) (*Cho et al., 2013*).

Viruses and viroids show a high level of genetic diversity in the infected host by replicating with strong mutation rates (*Sanjuan et al., 2010*). Therefore, viral populations in the host are composed of diverse variants of viruses and viroids, which are called viral quasispecies, rather than a single unique viral genome (*Domingo, Sheldon & Perales, 2012*). Viral quasispecies affect the genetic diversity and pathogenicity of viruses and viroids. The nature of viral quasispecies has been previously characterized in RNA and DNA viruses infecting plants (*Duffy & Holmes, 2008*; *Schneider & Roossinck, 2001*). In addition, viroids exhibit quasispecies. For example, the quasispecies of CSVd and CChMVd in the chrysanthemum host have been characterized (*Codoñer et al., 2006*). Using agrobacterium-mediated infiltration, evidence for quasispecies of CSVd in an infected single chrysanthemum plant has been demonstrated (*Nabeshima, Doi & Hosokawa, 2016*). Furthermore, genetic variations of CSVd in different chrysanthemum plants in Brazil (*Gobatto et al., 2014*) and *Argyranthemum frutescens* plants in Italy (*Torchetti et al., 2012*) have been reported. However, the genetic variations of CSVd in different chrysanthemum cultivars in Korea and the association of genetic variants with the host have not been well studied.

In this study, we analyzed the genetic variations of CSVd genomes in different chrysanthemum cultivars in Korea in order to elucidate the genetic diversity and quasispecies of CSVd by cloning-based Sanger sequencing.

## MATERIALS AND METHODS

### Plant samples

All chrysanthemum plants in this study were purchased from the Gangnam Flower Market, Seoul, on January 22, 2015. Leaf samples were harvested from a single plant for each cultivar and frozen immediately using liquid nitrogen. All frozen leaf samples were kept at $-80\,^{\circ}\text{C}$ for further experimentation.

### RNA isolation and RT-PCR

The frozen leaf samples were ground in liquid nitrogen with a mortar and pestle. The total RNAs were extracted using the RNeasy Plant Mini Kit (Qiagen, Hilden,

**Table 1 Detailed information on chrysanthemum plants used in this study.** Cultivar name, scientific name and cultivated geographical location are provided. In the case of the Shinma cultivar, the plants were obtained from two different locations: China (SMC) and Jeju, Korea (SMJ). From each plant, at least 20 clones were sequenced and their sequences were deposited in GenBank with their respective accession number. The number of variants in each plant is also indicated.

| Index | Abbreviation | Cultivar name | Scientific name | Location | No. of clones | No. of variants | Length | Accession no. |
|---|---|---|---|---|---|---|---|---|
| 1 | BRJ | Borami | *Dendranthema x grandiflora* | Jeju | 20 | 12 | 354 | KX096347–KX096366 |
| 2 | FDJ | Ford | *Dendranthema x grandiflora* | Jeju | 22 | 12 | 354 | KX096408–KX096428 |
| 3 | YSI | Yes Song | *Dendranthema x grandiflora* | Icheon | 22 | 8 | 354 | KX096555–KX096576 |
| 4 | YCI | Yellow Cap | *Dendranthema x grandiflora* | Icheon | 23 | 14 | 354 | KX096533–KX096554 |
| 5 | SMC | Shinma | *Dendranthema x grandiflora* | China | 24 | 8 | 354 | KX096472–KX096491 |
| 6 | DCI | Disk Club | *Dendranthema x grandiflora* | Icheon | 25 | 20 | 354 | KT005803–KT005827 |
| 7 | VCS | Vatican | *Dendranthema x grandiflora* | Siheung | 20 | 10 | 354 | KX096512–KX096532 |
| 8 | FGI | Froggy | *Dendranthema x grandiflora* | Icheon | 22 | 6 | 354 | KX096429–KX096449 |
| 9 | FPJ | Fire Pink | *Dendranthema x grandiflora* | Jeju | 24 | 14 | 354 | KX096388–KX096407 |
| 10 | PTG | Pink This Plus | *Dendranthema x grandiflora* | Gangneung | 23 | 15 | 354 | KX096450–KX096471 |
| 11 | ISJ | Isokuk | *Dendranthema pacificum* | Jeju | 23 | 14 | 354 | KX096367–KX096387 |
| 12 | SMJ | Shinma | *Dendranthema x grandiflora* | Jeju | 23 | 15 | 354 | KX096492–KX096511 |

Germany) according to the manufacturer's instructions. The extracted total RNAs were subjected to reverse transcription polymerase chain reaction (RT-PCR) using CSVd-specific primers: 5′-AAAGAAATGAGGCGAAGAAGTC-3′ (position 1–22) and 5′-TTCTTTCAAAGCAGCAGGGT-3′ (position 335–354) (*Choi et al., 2015*). RT-PCR was conducted using a DiaStar OneStep RT-PCR Kit composed of RTase and Solg h-Taq DNA polymerase according to the manufacturer's instructions (SolGent, Daejeon, Korea).

## Cloning and sequencing

The amplified PCR products from CSVd-infected plant samples were cloned using pGEM-T-Easy Vector (Promega, WI, USA). For each plant sample, at least 20 clones were subjected to Sanger sequencing. The obtained CSVd genome sequences were deposited in GenBank with their respective accession number. The accession numbers of the CSVd genome sequences for the individual chrysanthemum plants are listed in Table 1.

## Sequencing analysis and phylogenetic analysis

A total of 271 clones were sequenced. We analyzed all 271 CSVd genome sequences and identified 105 variants. To generate a consensus CSVd genome sequence, all 271 CSVd genome sequences were aligned by ClustalW implemented in the MEGA6 program with

default parameters (*Thompson, Gibson & Higgins, 2002*). After alignment, we calculated the percentage of each nucleotide among the 271 genomes, and the nucleotide with the highest percentage in each genome was used for the generation of a consensus CSVd genome sequence. In the same way, we generated a consensus CSVd genome sequence from each single plant. The 12 CSVd consensus genome sequences were subjected to the construction of a phylogenetic tree using the MEGA6 program (*Tamura et al., 2013*). For the construction of the phylogenetic tree, the genome sequences were aligned by ClustalW and the neighbor-joining method was employed. In the construction of the phylogenetic tree for the 105 CSVd variants, the aligned sequences were converted into the NEXUS file format using MEGA6, and then the NEXUS file was imported into the SplitsTree4 program (*Huson & Bryant, 2006*). Finally, the unrooted phylogenetic tree was constructed by SplitsTree4 using the neighbor-joining method.

## Analysis of single-nucleotide variations and recombination analysis

In order to examine the SNVs of the CSVd genome in a single chrysanthemum plant, we generated a consensus CSVd genome sequence from each plant. We aligned each genome sequence against the generated consensus CSVd genome sequence derived from each chrysanthemum plant. To identify the SNVs within the 105 identified variants, the sequences for the 105 variants were aligned against a consensus CSVd genome sequence that was obtained by combining all 105 variants. The aligned sequences for the 105 variants were subjected to recombination analysis using the RDP program with default parameters (*Martin et al., 2015*).

# RESULTS

## Identification of CSVd-infected chrysanthemum plants from commercial chrysanthemum cultivars

We sampled 36 different chrysanthemum cultivars from the market and examined the infection of CSVd in obtained chrysanthemum cultivars by RT-PCR using CSVd-specific primers. In this study, we did not consider the disease symptoms caused by CSVd. Out of the 36 cultivars, 11 cultivars were infected by CSVd. Among the CSVd-infected chrysanthemum cultivars, the Shinma cultivars were derived from two different regions: Jeju, Korea, and an unknown city in China. The 12 plant samples were named based on their cultivar name and cultivation region (Table 1 and Fig. 1A). Five cultivars were derived from Jeju, and four cultivars were grown in Incheon. Except for the Isokuk cultivar belonging to *Dendranthema pacificum*, all other cultivars were members of *Dendranthema x grandiflora*.

## Amplification of CSVd genomes from 12 different chrysanthemum plants

In order to examine the genetic variations of CSVd genome sequences in a single chrysanthemum plant, we conducted RT-PCR again using CSVd-specific primers and sequenced the amplified CSVd genomes by cloning-based Sanger sequencing. We only used the total RNAs extracted from a single plant representing an individual chrysanthemum cultivar except for the Shinma cultivar, which was further divided into SMJ and SMC based

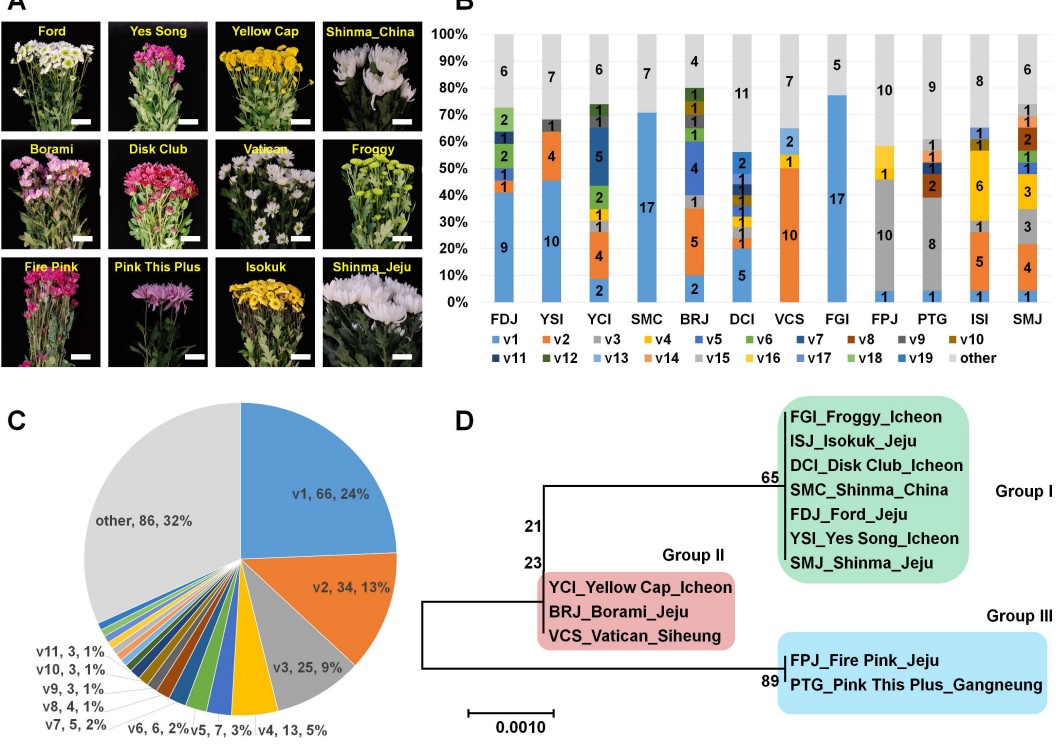

**Figure 1  Analysis of genetic variations for CSVd genomesderived from 12 chrysanthemum plants composed of 11 different cultivars.** (A) Images of 12 chrysanthemum plants infected by CSVd used for amplification of CSVd genome sequences. (B) Number of identified variants in individual chrysanthemum plant. (C) Distribution of identified CSVd variants based on number of clones in individual chrysanthemum plant. (D) The phylogenetic tree for 12 consensus CSVd genomes constructed using the neighbour-joining method with 1,000 bootstrap replicates.

on origin (Table 1). From each cultivar, at least 20 clones were sequenced, resulting in a total of 271 genome sequences (Table S1). Interestingly, the sizes of all sequenced CSVd genomes were identical at 354 (nt) in length, and the identified number of variants in each cultivar ranged from 6 to 20 (Table 1). The Froggy cultivar possessed only six variants, while the Disk Club cultivar displayed at least 20 variants.

## Comparison of identified CSVd variants in 12 chrysanthemum plants

We analyzed all 271 CSVd genome sequences obtained and identified 105 CSVd variants (Table S2). The dominant CSVd variant in each cultivar was variable (Fig. 1B). For example, CSVd variant 1 (v1) was dominant in the Ford, Yes Song, and Shinma cultivars from China and in the Disk Club and Froggy cultivars. In the Borami, Vatican, and Shinma cultivars from Jeju, v2 was dominant, whereas v3 was dominant in the Fire Pink and Pink This Plus cultivars.

We examined the number of sequenced CSVd genomes for each variant (Table S3 and Fig. 1C) and found that v1 (66 genomes) was dominant, followed by v2 (34 genomes), v3 (25 genomes), and v4 (13 genomes). Eleven variants contained more than two genomes.

Based on the number of each variant, v1 represented 24%, followed by v2 (13%), v3 (9%), and v4 (5%). The four major variants represented 51% of all CSVd variants.

## Phylogenetic relationships of identified CSVd genomes

We were interested in whether the cultivation region is an important factor for CSVd genome variation. Thus, we generated a phylogenetic tree using CSVd consensus genome sequences from 12 plants (Fig. 1D). The phylogenetic tree displayed three groups of CSVd genomes: Group I contained seven cultivars, including the Shinma cultivar from China and Jeju, Korea; Group II contained the Yellow Cap, Borami, and Vatican cultivars; and Group III had only two cultivars: Fire Pink and Pink This Plus.

We generated an unrooted phylogenetic tree for 105 CSVd variants using SplitsTree to reveal the phylogenetic relationships among the 105 CSVd variants (Fig. 2A), and the phylogenetic tree revealed at least six groups of CSVd variants. Group A was the largest, containing 24 variants, including v1 and v11, Group C was the second largest with 21 variants, including v3 and v8, and Group B was the third largest, possessing 18 variants, including v2. Groups D, E, and F contained 13, 6, and 8 variants, respectively. Some variants, such as v68 and v96, which belonged to Groups A and C, respectively, were distinct from other variants in the same group. The phylogenetic tree suggested that v1 to v6 were the main CSVd genomes in each group and that several other CSVd variants might have been derived from them.

## Generation of a consensus CSVd genome sequence and analysis of single-nucleotide variations

In order to examine SNVs for identified CSVd genomes, we generated a consensus CSVd genome combining all CSVd genome sequences. The consensus CSVd genome was identical to the known CSVd strain SK1 (accession number: AB679193.1). After the sequence alignment of all variants against the consensus, the CSVd genome sequence revealed 99 SNVs in the CSVd genome 354 nt in length (Table S4). In general, most nucleotides in the CSVd genome were highly conserved among the 105 variants in this study. However, three nucleotide positions showed high levels of sequence variations (Fig. 2B). The three SNVs were localized at 298 (A–U with 50.47%), 256 (C–U with 77.14%), and 156 (A–G with 80%). We further analyzed the recombinant events among the 105 CSVd variants using the RDP4 program, and no recombinants were detected.

## DISCUSSION

In this study, we examined genetic variations of the CSVd genome in single chrysanthemum plants. Thus, we investigated the infectivity of CSVd in different randomly selected chrysanthemum cultivars. Finally, CSVd genomes amplified from 12 plants from 11 cultivars were sequenced. As we expected, each chrysanthemum plant had a different set of CSVd variants. For instance, the compositions of the CSVd variants were different in the Shinma cultivar from Jeju, Korea and the Shinma cultivar from China, although the host was identical. Interestingly, the consensus CSVd genome sequences in the two Shinma plants were identical. This result showed that some nucleotides in the CSVd genome could

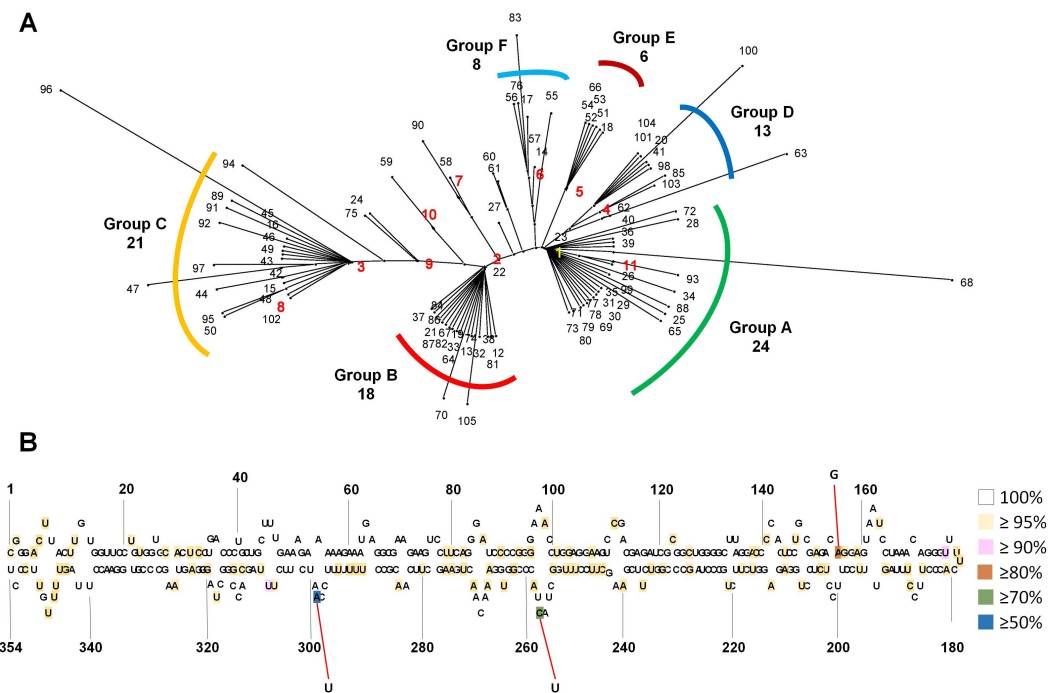

**Figure 2   Phylogenetic relationship and identification of single-nucleotide variations for 105 CSVd variants.** (A) The unrooted phylogenetic tree for 105 CSVd variants was constructed using the SplitsTree4 programme with the neighbour-joining method. Each number indicates an individual variant. The top ten variants are indicated by yellow and red with bold characters. (B) Identification of single-nucleotide variations in the consensus CSVd genome. The secondary structure of the CSVd genome was adapted from a previous study (*Yoon & Palukaitis, 2013*). Number indicates the position of the nucleotide. The three nucleotide positions showing high mutation rates are indicated by red lines.

be changed; however, the dominant CSVd genome sequence in a certain chrysanthemum cultivar might be highly conserved.

The phylogenetic tree using 12 consensus CSVd genomes revealed three groups of CSVd genomes. This result demonstrated that at least three dominant CSVd variants are present in chrysanthemum plants grown in Korea. In fact, the result on the phylogenetic relationship was highly correlated with the results on the composition of CSVd variants in the individual plants. For instance, the dominant variant in Group I was v1, while v2 and v3 were dominant in Groups II and III, respectively.

The unrooted phylogenetic tree using 105 CSVd variants revealed at least six groups of CSVd variants. The advantage of this phylogenetic tree is that it can show the dominant CSVd variant as well as its sister variants derived from the dominant CSVd variant. For example, among the 24 CSVd variants in Group A, v1 was the dominant CSVd genome, while the other 23 variants were derived from v1.

Interestingly, the consensus CSVd genome sequences as well as v1 were identical to the previously identified CSVd SK1 genome (*Yoon et al., 2014*). In addition, v2 and v3 displayed only one nucleotide and three nucleotide substitutions compared with CSVd SK1. SK1 was previously identified in a chrysanthemum showing stunt disease symptoms, and it was also used for the construction of an infectious clone for CSVd

(*Yoon et al., 2014*). Therefore, the majority of CSVd-infected chrysanthemum plants in this study might be symptomatic, and the potential risk of CSVd in chrysanthemum production in Korea might be very high.

According to a previous study, there are genome sequences for at least 117 CSVd isolates worldwide, and only 24 isolates divided into six variants have been sequenced in Korea (*Yoon & Palukaitis, 2013*). In this study, we sequenced 271 CSVd genomes, enough to obtain the genetic variations of CSVd genomes in Korea. Our results suggest that the number of identified CSVd variants could be increased as the number of sequenced genomes is increased. However, the consensus genome sequences for CSVd in Korea could be identical, suggesting the conservancy of the CSVd genome.

Based on a consensus CSVd genome sequence, 99 SNVs were identified from 105 variants; however, no indel (insertion and deletion) was identified. The 99 SNVs were scattered through the CSVd genome. This result is consistent with a previous study that analyzed all available CSVd genome sequences (*Yoon & Palukaitis, 2013*). However, the mutation rates for most SNVs were very low, suggesting the high conservancy level of the CSVd genome, which is different from CChMVd, which displayed high mutation rates (*Gago et al., 2009*). We found three nucleotide positions that displayed strong mutation rates among the 105 variants. A previous study identified seven positions (47, 49, 50, 64, 65, 254, and 298) that showed strong sequence variations among 80 examined variants of CSVd (*Yoon & Palukaitis, 2013*). Interestingly, position 298 was identified by our study. At position 294, v1, v4, v5, and v6 had A, while v2, v3, v7, v8, v9, and v10 had U. Although specific sequences in the tetraloop of CChMVd associated with pathogenicity have been revealed (*De la Peña, Navarro & Flores, 1999*), the specific nucleotides or regions for the pathogenicity of CSVd have not been determined. Thus, it might be interesting to examine the functional roles of the 294 nucleotide associated with the CSVd lifecycle, such as replication, pathogenicity, and movement, in the near future.

Taken together, we revealed the quasispecies of CSVd in single chrysanthemum plants showing different genetic variations of the CSVd sequence in diverse chrysanthemum cultivars. Although 105 variants and 99 SNVs were identified from 271 CSVd genomes, the CSVd genomes in chrysanthemum plants grown in Korea were highly conserved.

## ACKNOWLEDGEMENTS

This work is dedicated to the memory of my father, Tae Jin Cho (1946–2015).

### Funding

This work was partially supported by the National Research Foundation of Korea (NRF) grant funded by the Korea government Ministry of Education (No. NRF-2016R1D1A1A02937216) and the support of the ''Next-Generation BioGreen21 Program (Project No. PJ01130902)'' of the Rural Development Administration, Republic of Korea. The funders had no role in study design, data collection and analysis, decision to publish, or preparation of the manuscript.

## Grant Disclosures

The following grant information was disclosed by the authors:

National Research Foundation of Korea (NRF).

Korea government Ministry of Education: NRF-2016R1D1A1A02937216.

Next-Generation BioGreen21 Program of the Rural Development Administration, Republic of Korea: PJ01130902.

## Competing Interests

The authors declare there are no competing interests.

## Author Contributions

- Hoseong Choi and Yeonhwa Jo conceived and designed the experiments, performed the experiments, analyzed the data.
- Ju-Yeon Yoon and Seung-Kook Choi conceived and designed the experiments, contributed reagents/materials/analysis tools, wrote the paper, prepared figures and/or tables, reviewed drafts of the paper.
- Won Kyong Cho conceived and designed the experiments, analyzed the data, contributed reagents/materials/analysis tools, wrote the paper, prepared figures and/or tables, reviewed drafts of the paper.

## DNA Deposition

The following information was supplied regarding the deposition of DNA sequences:

GenBank

KX096347–KX096366

KX096408–KX096428

KX096555–KX096576

KX096533–KX096554

KX096472–KX096491

KT005803–KT005827

KX096512–KX096532

KX096429–KX096449

KX096388–KX096407

KX096450–KX096471

KX096367–KX096387.

## Data Availability

The raw data has been supplied as a Supplemental Information 1.

## Supplemental Information

Supplemental information for this article can be found online at http://dx.doi.org/10.7717/peerj.2933#supplemental-information.

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
