# Peer review of "Sequence variability of Chrysanthemum stunt viroid in different chrysanthemum cultivars"

_PeerJ, doi:10.7717/peerj.2933_

## Round 0.1 · original submission · Major Revisions

Thank you for giving us the opportunity to consider your manuscript. Your paper has been reviewed by two referees who have raised important criticisms, which are detailed below. We will therefore consider publishing your manuscript if you can accommodate their suggestions in a revised version or explain satisfactorily why their comments are invalid.

·

Basic reporting

The authors of this manuscript performed an analysis of Chrysanthemum stunt viroid (CSVd) sequence variability in different chrysanthemum cultivars. This analysis is quite interesting and will contribute to better understand the biology of this viroid species. However, authors should re-write some parts of the manuscript to make it more realistic. My main concern is about the reiterative use of the word quasispecies, which is also included in the title. Although it is sensible to consider that when a viroid infects a plant its progeny behaves as a quasispecies, I recommend authors to avoid this term, which has more complex implications according to Eigen’s model, and in most parts of the manuscript write better: sequence variants or sequence variability. In methods, some questions need to be clarified. Finally, in discussion, some conclusions are unjustified and they should be removed.

Experimental design

Experimental design is fine. However, in cloning and sequencing (line 86), authors must inform about the enzymes used in the RT-PCR amplification. In a sequence variability analysis, it is important to know whether error prone polymerases or high-fidelity polymerases were used. Moreover, details of how the consensus sequences were built are needed (line 95). Authors should also emphasize where their amplification primers map in the viroid sequence and the implications about the experimental sequence variability in the region covered by the primers.

Validity of the findings

Findings seem valid.

Additional comments

These are some suggestions to improve the manuscript:

1. Change the title, replacing quasispecies by sequence variability.

2. Last sentence of the abstract must be absolutely removed.

3. Line 122: how many plants of each cultivar were tested positive for CSVd? Include a table with this relevant data. Data not shown is not a good idea in this particular case. Also the “at least” wording is not appropriate. If you had problems diagnosing plants; explain these problems.

4. Lines 189-190: remove this sentence. There is nothing in this study that suggests CSVd sequence variability arises from host genetic variability.

5. Lines 190-192: I also recommend deleting this conclusion. Control experiments are needed to connect sequence variability with environmental conditions. Variability of CSVd amplified from plants grown in different regions may also result from historical contingencies.

6. Lines 215-217: check this sentence. The pathogen is CSVd, not the infected plants.

Reviewer 2 ·

Basic reporting

In this work, authors have sequenced distinct isolates of Chrysanthemum stunt viroid (CSVd) from a variety of chrysanthemum cultivars from Korea (except one from China). Authors claim that this is the first report providing evidence of the quasispecies of CSVd in a single chrysanthemum plant as well as different cultivars (page 2, lines 20-21) and, indeed, assessment of that issue seems to have been the main objective of the work (page 4, Iines 13-14). However, evidence for quasispecies nature of CSVd populations has been previously obtained (Nabeshima et al., 2015, J Virol Methods ; Torchetti et al., 2012, J Plant Pathol 94) and interpopulation variation has been also reported somewhere else (Yoo and Palukaitis 2013, Virus Genes 46; Gobatto et al., 2014, J Plant Pathol 96). In that sense, I am not sure that there was any knowledge gap to fill in and I do not feel results are adding much to the previous knowledge on this issue. The work could have benefited, for instance, from the inclusion of information on disease symptoms and potential correlations with genetic population structure. Moreover, discussion is too long, not very well structured and some conclusions are not clearly supported by the data. In this sense, it is not clear at all that genetic variation is affected by the cultivar or by the cultivation region, particularly taking into account that a few variants are very common and the predominant ones in many cultivars. Moreover, clusterings shown in phylogenetic trees seem to contradict those potential conclusions, as clades do not correlate with either geographic origin or cultivars (Fig. 1D and 2A). I would suggest revision of the whole text and avoidance of unnecessary speculation. On the other side, all the manuscript needs to be carefully revised for language editing, there are grammatical errors, redundancies and very inaccurate expressions.

Experimental design

The methodological approach is standard (RT-PCR, cloning and sequencing of viroid variants) as is the analysis of the obtained sequences. In page 7, positions of RT-PCR primers on CSVd genome should be indicated. Moreover, the primers align on a CSVd region that might present sequence variation and corroboration of this issue should be done with another pair of primers.

Validity of the findings

See comments above

---

## Round 0.2 · accepted · Accept

The revised version of the manuscript has satisfactorily addressed the reviewer´s comments.